# Comparison of the Accuracy of Epistasis and Haplotype Models for Genomic Prediction of Seven Human Phenotypes

**DOI:** 10.3390/biom13101478

**Published:** 2023-10-03

**Authors:** Zuoxiang Liang, Dzianis Prakapenka, Yang Da

**Affiliations:** Department of Animal Science, University of Minnesota, Saint Paul, MN 55108, USA; zliang@umn.edu (Z.L.); praka032@umn.edu (D.P.)

**Keywords:** genomic prediction, epistasis, haplotype, prediction accuracy, SNP, human phenotype

## Abstract

The accuracy of predicting seven human phenotypes of 3657–7564 individuals using global epistasis effects was evaluated and compared to the accuracy of haplotype genomic prediction using 380,705 SNPs and 10-fold cross-validation studies. The seven human phenotypes were the normality transformed high density lipoproteins (HDL), low density lipoproteins (LDL), total cholesterol (TC), triglycerides (TG), weight (WT), and the original phenotypic observations of height (HTo) and body mass index (BMIo). Fourth-order epistasis effects virtually had no contribution to the phenotypic variances, and third-order epistasis effects did not affect the prediction accuracy. Without haplotype effects in the prediction model, pairwise epistasis effects improved the prediction accuracy over the SNP models for six traits, with accuracy increases of 2.41%, 3.85%, 0.70%, 0.97%, 0.62% and 0.93% for HDL, LDL, TC, HTo, WT and BMIo respectively. However, none of the epistasis models had higher prediction accuracy than the haplotype models we previously reported. The epistasis model for TG decreased the prediction accuracy by 2.35% relative to the accuracy of the SNP model. The integrated models with epistasis and haplotype effects had slightly higher prediction accuracy than the haplotype models for two traits, HDL and BMIo. These two traits were the only traits where additive × dominance effects increased the prediction accuracy. These results indicated that haplotype effects containing local high-order epistasis effects had a tendency to be more important than global pairwise epistasis effects for the seven human phenotypes, and that the genetic mechanism of HDL and BMIo was more complex than that of the other traits.

## 1. Introduction

Epistasis effects are gene interaction effects and global epistasis effects of single nucleotide polymorphism (SNP) markers are interaction effects of genome-wide SNPs. Several studies showed that global epistasis effects increased the accuracy of genomic prediction [1,2,3,4]. Haplotype effects may contain local high-order epistasis effects beyond pairwise epistasis effects and improved the prediction accuracy of genomic prediction in some cases [5,6,7,8,9,10]. For the Framingham Heart Study (FHS) data in this study, we previously reported that haplotypes improved the accuracy of predicting the phenotypic values of seven human phenotypes, with accuracy increases of 1.86–8.12% due to the use of haplotypes relative to the best SNP models, and the haplotype epistasis heritability showed that the epistasis effects contained in the haplotypes were responsible for the accuracy increases due to haplotypes [8]. The epistasis effects contained in the haplotypes were local high-order epistasis effects beyond pairwise epistasis effects because the average number of SNPs in haplotypes for the best haplotype prediction model was mostly more than 12 SNPs per haplotype block for the seven human phenotypes, although the exact number of SNPs involved in those high-order epistasis effects were unknown. Local high-order epistasis effects have the limitation of covering proximal SNPs only in a chromosome region or gene and cannot study interactions between distal variants on the same chromosome or variants on different chromosomes. In contrast, global epistasis effects can be used to study interaction effects between genetic variants anywhere on the genome, and potentially could utilize epistasis effects not covered by haplotypes for improving the accuracy of predicting phenotypic values. Similarly, local high-order epistasis effects could cover epistasis effects that are not feasible to study using the approach for global epistasis effects, as shown by the results of third- and fourth-order epistasis effects in this study. However, the comparison between local and global epistasis effects for their contributions to the phenotypic variance and the accuracy of genomic prediction was unavailable. We hypothesized that local epistasis can be more important than global epistasis for some traits, the reverse is true for some other traits, and some traits may involve both local and global epistasis effects. To test those hypothesis, and to discover and utilize the complex genetic mechanisms associated with those hypothesis, we developed a multifactorial method that integrates haplotype effects that contain local epistasis effects with global epistasis effects for genomic estimation and prediction [11] and implemented this method by the EPIHAP computer program [2]. In this study, we investigated the relationship between the accuracy of genomic prediction of seven human phenotypes and global low-order epistasis effects as well as local high-order epistasis effects contained in haplotypes to answer two questions: whether global epistasis effects or haplotype effects were more important for predicting the seven human phenotypes, and whether the integration of global epistasis and haplotype effects would improve the accuracy of genomic prediction for any of the seven phenotypes using the multifactorial method implemented in the EPIHAP program.

## 2. Materials and Methods

### 2.1. Phenotypic and SNP Data

The Framingham Heart Study (FHS) data (2019 version) had 7565 individuals with genotypes of the 500 K SNP panel that had 488,146 autosomal SNPs. The seven phenotypes analyzed in this study were the normality transformed high density lipoproteins (HDL), low density lipoproteins (LDL), total cholesterol (TC), triglycerides (TG), weight (WT) using the Box-Cox transformation; and the original phenotypic observations without normality transformation of height (HTo) and body mass index (BMIo), with 3657–7564 observations, as previously described [8]. The SNP coordinates were converted to those of GRCh38.p13, and 486,356 SNPs had known chromosome positions on GRCh38.p13. From these 486,356 autosome SNPs, we previously studied eight SNP sets including the 380 K SNP set with 380,705 SNPs requiring minor allele frequency of 0.05, and this 380 SNP set was used in this study for epistasis and haplotype analysis. The haplotype blocking method for each trait was that with the highest prediction accuracy among over 140 haplotype models using structural and function genomic information in our previous study [8], i.e., the gene-based blocks for HDL, 12 SNPs per block for LDL, the 50 Kb blocks for TG, the 200 Kb blocks for HTo, 12 SNPs per block for WT, and the 100 Kb blocks for BMIo.

### 2.2. Mixed Model for GBLUP and GREML

Genomic best linear unbiased prediction (GBLUP) of genetic values was used for calculating the accuracy of predicting the phenotypic values, and genomic restricted maximum likelihood estimation (GREML) was used for estimating the heritability of each type of genetic effects using a multifactorial mixed model. The effect types evaluated by GBLUP and GREML included SNP additive and dominance effects, haplotype additive effects, and global epistasis effects up to the fourth-order. Details of the multifactorial model and the GBLUP and GREML based on this model are available in our methodology article [11]. The multifactorial mixed model can be briefly described as:(1)$$y=Xb+Zg+e=Xb+Z\sum_{i=1}^{f}u_{i}+e$$
(2)$$V=ZGZ^{\prime }+\;\sigma_{e}^{2}I_{N}=Z(\sum_{i=1}^{f}G_{i})Z^{\prime }+\;\sigma_{e}^{2}I_{N}=Z(\sum_{i=1}^{f}\sigma_{i}^{2}S_{i})Z^{\prime }+\;\sigma_{e}^{2}I_{N}$$
(3)$$G=\mathrm{Var}(\sum_{i=1}^{f}u_{i})=\sum_{i=1}^{f}G_{i}=\sum_{i=1}^{f}\sigma_{i}^{2}S_{i}$$
where **y** = N × 1 column vector of phenotypic observations, **Z** = N × n incidence matrix allocating phenotypic observations to each individual = identity matrix for one observation per individual (N = n), N = number of observations, n = number of individuals, **b** = c × 1 column vector of fixed effects, c = number of fixed effects, **X** = N × c model matrix of **b**, **e** = N × 1 column vector of random residuals, $\sigma_{e}^{2}$ = residual variance, $u_{i}$ = n × 1 column vector of the genetic effects of the $i_{}^{\mathrm{th}}$ effect type, $\sigma_{i}^{2}$ = variance of the genetic effects of the $i_{}^{\mathrm{th}}$ effect type, $S_{i}$ = n × n genomic relationship matrix of the $i_{}^{\mathrm{th}}$ effect type, and f = 15 in this study. Subscript i=1−14 in Equations (1) and (2) represent SNP and epistasis effects (Table 1), and subscript i=15 represents haplotype additive effect. Fixed effects included sex and cholesterol treatment as classification variables, and age, glucose and BMIo as covariables for HDL, TC, LDL and TG; and sex as classification variable and age as covariable for HTo, WT and BMIo. The calculations of the genomic epistasis relationship matrices ($S_{i}$) can use either the approximate method that is the genomic version of Henderson’s Hadamard products between additive and dominance relationship matrices [12,13,14,15] or the exact method that removes intra-locus epistasis that should not exist from the approximate method [16]. The EPIHAP package implemented both methods, and this study used the exact method although the approximate and exact methods had similar results for a swine dataset [11] and the same results for a Holstein dataset [2].

### 2.3. Evaluation of Prediction Accuracy Using Cross-Validation

A 10-fold validation study was used to evaluate the accuracy of predicting the phenotypic values of each trait for each model. Individuals with phenotypic observations were randomly divided into 10 validation populations. Each validation poluation was predicted by the training population that consists of all the remaining nine validation populations. The first nine validation populations had equal sample size each and the tenth population had the remaining individuals. In each validation population, phenotypic values were omitted when calculating GBLUP for training and validation individuals. The observed accuracy of predicting the phenotypic values (predictive ability, [17]) was defined as the correlation between GBLUP of the genotypic values and the phenotypic values in each validation population and then averaged over all validation populations, i.e.,:(4)$${\hat{R}}_{0p}=\mathrm{corr}({\hat{g}}_{0},\;y_{0})=[\sum_{k=1}^{10}\mathrm{corr}({\hat{g}}_{0k},\;y_{0k})]/10$$
where ${\hat{g}}_{0}$ = GBLUP of $g_{0}^{}$; $g_{0}^{}$ = unobservable genetic values; $y_{0}^{}$ = phenotypic observations; subscript ‘0’ denotes validation population; ‘corr’ stands for correlation.

### 2.4. Initial Selection of Epistasis Effects for Prediction Models

The purpose of the initial selection of epistasis models was to exclude effect types with little or no contribution to the phenotypic variance from further evaluation for prediction accuracy. The initial model selection required each effect type to have a heritability estimate greater than 1% for the effect type to be included in the prediction model for further evaluation.

## 3. Results and Discussion

### 3.1. Initial Epistasis Models for Predicting Phenotypic Values

Heritability estimates using the model with SNP and epistasis effects up to the fourth-order showed that the fourth-order virtually had no contribution to the phenotypic variances of the seven phenotypes (Table 1). Among the 35 heritability estimates of the fourth-order epistasis effects for seven traits, only six were nonzero in the range of 0.001–0.004, with five nonzero estimates for WT and one for TC. Since none of the heritability estimates of the fourth-order epistasis effects reached the 0.01 threshold we required, the fourth-order epistasis effects were excluded from the prediction model for any trait. For four traits (LDL, TC, TG and HTo), A × A was the only epistasis effect with heritability greater than the 0.01 threshold, ranging from 0.135 for TG to 0.423 for LDL. Therefore, the prediction model for these four traits was A + D + AA. HDL and BMIo had A × D heritability estimates but no third-order epistasis heritability estimates exceeded the 0.01 threshold value, and the initial prediction model for these two traits were the A + D + AA + AD models. For the third-order epistasis effects, only WT had heritability estimates greater than the 0.01 threshold, 0.017 for A × A × A and 0.014 for A × A × D, and the initial prediction model was A + D + AA + AD + DD + AAA + AAD (Table 1).

**Table 1 biomolecules-13-01478-t001:** Heritability estimates of the full model with SNP and epistasis effects up to the fourth-order.

	Subscript in Equations (1) and (2)	HDL	LDL	TC	TG	HTo	WT	BMIo
A	1	*0.241*	*0.284*	*0.331*	*0.221*	*0.648*	*0.424*	*0.320*
D	2	*0.055*	*0.034*	*0.085*	*0.079*	*0.134*	*0.071*	*0.012*
A × A	3	*0.356*	*0.423*	*0.145*	*0.135*	*0.192*	*0.106*	*0.219*
A × D	4	*0.111*	0.000	0.003	0.000	0.000	*0.057*	*0.126*
D × D	5	0.001	0.000	0.001	0.000	0.000	*0.015*	0.008
A × A × A	6	0.002	0.000	0.004	0.000	0.000	*0.017*	0.000
A × A × D	7	0.002	0.000	0.001	0.000	0.000	*0.014*	0.000
A × D × D	8	0.000	0.000	0.001	0.000	0.000	0.008	0.000
D × D × D	9	0.000	0.000	0.000	0.000	0.000	0.004	0.000
A × A × A × A	10	0.000	0.000	0.001	0.000	0.000	0.004	0.000
A × A × A × D	11	0.000	0.000	0.000	0.000	0.000	0.003	0.000
A × A × D × D	12	0.000	0.000	0.000	0.000	0.000	0.002	0.000
A × D × D × D	13	0.000	0.000	0.000	0.000	0.000	0.001	0.000
D × D × D × D	14	0.000	0.000	0.000	0.000	0.000	0.001	0.000
Total heritability		0.768	0.742	0.573	0.437	0.974	0.728	0.686
Initial model		A + D + AA + AD	A + D + AA	A + D + AA	A + D + AA	A + D + AA	A + D + AA + AD + DD + AAA + AAD	A + D + AA + AD

A is additive effect. D is dominance effect. A × A, A × D and D × D are second-order (pairwise) epistasis effects. A × A × A, A × A × D, A × D × D and D × D × D are third-order epistasis effects. A × A × A × A, A × A × A × D, A × A × D × D, A × D × D × D and D × D × D × D are fourth-order epistasis effects. Entries in italic are heritability estimates greater than 0.01 for the initial prediction models. HDL is the normality transformed high density lipoproteins. LDL is the normality transformed low density lipoproteins. TC is the normality transformed total cholesterol. TG is the normality transformed triglycerides. WT is the normality transformed weight (WT). HTo is the original phenotypic observations without normality transformation of height, BMIo is the original phenotypic observations without normality transformation of body mass index.

### 3.2. Accuracy of Epistasis Models for Predicting Phenotypic Values

The comparison of seven models with epistasis effects for WT showed the model with A × A × A and A × A × D had the same observed prediction accuracy as the five models with pairwise epistasis effects with observed prediction accuracy of 0.325, a 0.62% increase over the A + D SNP model (Appendix A). These results indicated that A × A × A and A × A × D effects had no contribution to the observed accuracy of predicting WT. Among the five models with pairwise epistasis effects for WT, the A + AD and A + AD + DD models had the same prediction accuracy of 0.325, indicating D × D had no contribution to the prediction accuracy of WT. Therefore, the A + D + AD model was the best prediction model for WT with the smallest number of effect types. For BMIo, the A + AD and A + AD + DD models had the highest observed prediction accuracy of 0.325, indicating that D × D effects had the same contribution to the observed prediction accuracy of BMIo as the A × D (Appendix A). Therefore, either A × D or D × D could be included in the model but including both were unnecessary, and we used the A + AD mode for BMIo. For HDL, the A + D + AA + AD model had the highest prediction accuracy of 0.297 (Appendix A) and was the best epistasis model. Among the seven traits, HDL, WT and BMIo were the only three traits that had A × D effects in the final prediction models, and WT and BMIo were the only two traits that did not have A × A effects in the prediction models. These prediction models with A×D effects indicated that the genetic mechanism of HDL, WT and BMIo likely was more complex than that of LDL, TC and HTo.

The final epistasis models with the highest prediction accuracy and smallest number of effect types from 10-fold cross-validations for the seven traits are summarized in Table 2 and Figure 1. Of these seven traits, epistasis effects achieved accuracy increases over the SNP model for predicting the phenotypic values for six of the seven traits. Among these seven traits, LDL and HDL had the largest accuracy increases of 3.85% and 2.41% respectively, and four traits had minor accuracy increases for four traits, TC, HTo, WT and BMIo, with accuracy increases of 0.70%, 0.97%, 0.62% and 0.93% respectively. However, epistasis effects (A × A) decreased the prediction accuracy for TG (−2.35%) for unknown reasons. Compared to the observed prediction accuracy of the best haplotype models, none of the epistasis models reached the prediction accuracy of the best haplotype models (Table 2, Figure 1), showing that epistasis effects were not as accurate as haplotype effects for predicting the seven human phenotypes, or, local high-order epistasis effects contained in haplotypes were more important than global low-order epistasis effects. This conclusion was further confirmed by the results of the integrated prediction models with both epistasis and haplotype effects.

### 3.3. Accuracy and Heritability Estimates of Integrated Models with Epistasis and Haplotype Effects

The integrated model for each trait includes the haplotypes of the best haplotype blocking methods identified from our previous study [8] and the best epistasis model identified in this study. The results showed that the integration of epistasis and haplotype effects slightly increased the prediction accuracy over the best haplotype model for two traits, HDL (3.10% accuracy increase of the integrated model versus 2.76% accuracy increase of the best haplotype model), and BMIo (2.48% versus 2.17%), but did not improve the prediction accuracy of the haplotype models for five traits, LDL (7.69% versus 8.12%), TC (3.15% versus 3.15%), TG (0% versus 3.29%), HTo (2.18% versus 2.18%), and WT (1.86% versus 1.86%) (Table 2). Haplotype additive effects almost accounted for all the SNP additive heritability for three traits, with additive heritability reduced to 0.01 from 0.286 for LDL, from 0.327 for TC and from 0.460 for WT, whereas the haplotype additive heritability was 0.517 for LDL, 0.403 for TC, and 0.427 for WT (Table 3, or Appendix A with standard deviations). Haplotype additive effects also accounted for most of the A × A heritability for the five traits in the integrated model, with A × A heritability reduced to 0.042, 0.134, 0.017, 0.082 and 0.001 from 0.358, 0.429. 0.162, 0.171 and 0.183 for HDL, LDL, TC, TG and HTo, respectively (Table 3). These results of integrated models further confirmed that local high-order epistasis effects in haplotypes were more important than global pairwise epistasis effects for all seven human phenotypes in this study.

## 4. Conclusions

Heritability estimates and the accuracy of genomic prediction of seven human phenotypes showed that global fourth-order epistasis effects virtually had no contribution to the phenotypic variances, and global third-order epistasis effects did not affect the prediction accuracy. Global low-order (pairwise) epistasis effects improved the accuracy of predicting six of the seven phenotypes but not as much as the accuracy of haplotype effects. The integrated models with global low-order (pairwise) epistasis effect and haplotype effects had slightly higher prediction accuracy than the haplotype models for two traits whereas the haplotype models had the better prediction accuracy for the remaining five traits. These results indicated that local high-order epistasis effects in haplotypes had a tendency to be more important than global pairwise epistasis effects for the seven human phenotypes.

## Figures and Tables

**Figure 1 biomolecules-13-01478-f001:**
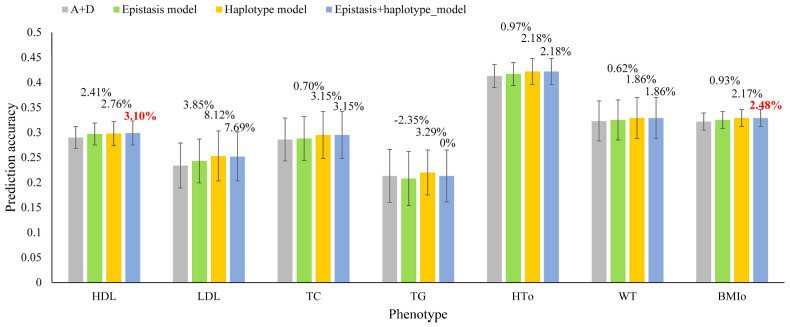
Accuracy of four prediction models for predicting phenotypic values of seven human phenotypes from 10-fold cross-validation studies. The accuracy measure was the correlation between the GBLUP genetic values and the phenotypic values in each validation population and then averaged over all validation populations (Equation (4)). Each error bar was one standard deviation of the prediction accuracies above and below each average.

**Table 2 biomolecules-13-01478-t002:** Accuracy of different models for predicting the phenotypic values in the validation populations.

	HDL	LDL	TC	TG	HT_O_	WT	BMI_O_
	SNP model (A + D)						
Prediction accuracy	0.290	0.234	0.286	0.213	0.413	0.323	0.322
	Global epistasis model						
Prediction model	A + D + AA + AD	A + D + AA	A + D + AA	A + D + AA	A + D + AA	A + D + AD	A + AD
Prediction accuracy	0.297	0.243	0.288	0.208	0.417	0.325	0.325
Accuracy increase over SNP model (%)	2.41	3.85	0.70	−2.35	0.97	0.62	0.93
	Haplotype model [8]						
Prediction model	A + D + H	H	D + H	H	A + D + H	A + D + H	A + H
Haplotype blocking method	Genes	12 SNPs	50 Kb	50 Kb	200 Kb	12 SNPs	100 Kb
Prediction accuracy	0.298	0.253	0.295	0.220	0.422	0.329	0.329
Accuracy increase over SNP model (%)	2.76	8.12	3.15	3.29	2.18	1.86	2.17
	Integrated model with epistasis and haplotype effects						
Prediction model	A + D + AA + AD + H	A + D + AA + H	A + D + AA + H	A + D + AA + H	A + D + AA + H	A + D + AD + H	A + AD + H
Prediction accuracy	0.299	0.252	0.295	0.213	0.422	0.329	0.330
Accuracy increase over SNP model (%)	3.10	7.69	3.15	0	2.18	1.86	2.48

Prediction accuracy is observed accuracy of predicting phenotypic values (Equation (4)). A is SNP additive values. D is SNP dominance values. H is haplotype additive values. AA is A×A values. AD is A×D values.

**Table 3 biomolecules-13-01478-t003:** Genomic heritability estimates as averages from 10-fold validations.

	HDL	LDL	TC	TG	HTo	WT	BMIo
	SNP model (A + D)						
A	0.386	0.408	0.389	0.260	0.740	0.472	0.415
D	0.124	0.177	0.104	0.124	0.202	0.093	0.046
Total	0.510	0.585	0.493	0.383	0.942	0.565	0.462
	Global epistasis model						
Model	A + D + AA + AD	A + D + AA	A + D + AA	A + D + AA	A + D + AA	A + D + AD	A + AD
A	0.243	0.286	0.327	0.211	0.654	0.460	0.398
D	0.056	0.037	0.087	0.069	0.137	0.064	-
A × A	0.358	0.429	0.162	0.171	0.183	-	-
A × D	0.115	-	-	-	-	0.188	0.265
Total	0.771	0.752	0.576	0.451	0.975	0.712	0.663
	Integrated model with global epistasis effects and haplotype effects						
Model	A + D + AA + AD + H	A + D + AA + H	A + D + AA + H	A + D + AA + H	A + D + AA + H	A + D + AD + H	A + AD + H
A	0.102	0.010	0.010	0.072	0.361	0.010	0.119
D	0.041	0.040	0.078	0.062	0.145	0.043	-
A × A	0.042	0.134	0.017	0.082	0.001	-	-
A × D	0.264	-	-	-	-	0.105	0.135
H	0.332	0.517	0.403	0.206	0.494	0.427	0.356
Total	0.780	0.702	0.538	0.421	1.0	0.687	0.61

A is SNP additive values. D is SNP dominance values. H is haplotype additive values. AA is A × A values. AD is A × D values.

## Data Availability

The datasets presented in this study are available from dbGaP, and data access requires approval from the National Heart, Lung, and Blood Institute (NHLBI).

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
