# Peer review of "Comparison of the Accuracy of Epistasis and Haplotype Models for Genomic Prediction of Seven Human Phenotypes"

_biomolecules, 2023, doi:10.3390/biom13101478_

Round 1
Reviewer 1 Report
In this study, the authors compared the variance components (as well as the genomic heritability estimates) and genome-wide prediction accuracy of haplotype models (representing local epistatic effects, both low- and high-order, within the haplotype block) and global low-order epistasis models on seven traits in a human data set. It was found that for this data set, the local high-order epistatic effects contributed more than the global low-order epistatic effects to the genetic variance. The following are some comments and suggestions which may help the authors to improve the quality of the paper.
Line 23: It seems that the sentence was not completed.
Line 71: Which method was used for the transformation?
Line 120-123: I do know that many studies in the literature implemented cross-validation in this way. And it has little influence on the prediction accuracy when sample size is large (which is the case for this study). Nevertheless, I would like to comment that a more proper approach of cross-validation is to pool the predicted values in each fold together and then calculate the correlation between the predicted and observed values once for the entire data set. The dividing and predicting procedure can be repeated.
Line 125: Equation (4) was not correctly displayed. “g_0” should be “g_0^hat”. There is a Chinese character on top of the equation. In addition, please use “cor” instead of “corr” for correlation.
Line 139-141: There are several errors in this statement. First, 5 different types of fourth-order epistasis times 7 traits give 35 heritability estimates, not 28. Second, I only observed 6 non-zero estimates ranged from 0.001 to 0.004, in which 5 for WT and one for TC.
Line 166: First of all, there were nine models (not seven) evaluated in Table S1. Secondly, what I understood is that the initial model listed in Table 1 was used as an “upper bound”. But still, not every possible combination smaller than the initial model was evaluated (e.g. A+D+AA+AD+DD+AAA). So how did you determine the models for evaluation?
Line 170-173: But both A+AD and A+AD+DD were not listed in Table 1. On the other hand, the accuracy of A+D+AA is 0.324, and A+D+AA+DD is 0.325. So, it seems that one cannot conclude that DD has no contribution.
Line 174-176: For BMIo, the model A+D+AA+AD+DD was evaluated (Table S2), but it exceeds the corresponding initial model listed in Table 1, which is A+D+AA+AD. So, the same question as for line 166: how did you determine the models for evaluation? And the same for the trait HDL.
Line 179-183: How was the best prediction models for the other four traits determined? In Tables S1-S3, you only showed the results of three traits.
Line 186-188: I would not call it a “substantial increase”. Even for the trait LDL with the largest relative increase (3.85%), the absolute increase of the mean accuracy was less than 0.01.
In Table 2 and Figure 1, I would suggest write “global epistasis model” instead of “epistasis model”.
Line 244-246: The difference between the prediction accuracies was actually quite small, which I think is not enough to make such a strong conclusion. On the other hand, the heritability estimates in Table 3 indeed indicated that the local high-order epistasis contributed more than the global low-order epistasis to the genetic variance. There is still one point which has to be considered: The covariance matrices for the different types of genetic effects could be correlated. Thus, the relative contribution of different types of genetic effects may not be reliably inferred from the variance components (as well as the heritability estimates).
Author Response
We thank the reviewer for a very helpful review.
Texts in yellow are in response to the reviewer comments, and texts in green are previous texts answering some reviewer questions.
Line 23: It seems that the sentence was not completed.
ANSWER: Corrected. Thanks.
Line 71: Which method was used for the transformation?
Line 120-123: I do know that many studies in the literature implemented cross-validation in this way. And it has little influence on the prediction accuracy when sample size is large (which is the case for this study). Nevertheless, I would like to comment that a more proper approach of cross-validation is to pool the predicted values in each fold together and then calculate the correlation between the predicted and observed values once for the entire data set. The dividing and predicting procedure can be repeated.
ANSWER: The validation study in this article is exactly what the reviewer describes: 9 folds predict 1 fold. A sentence is added to reflect this fact. However, each fold has a chance to the validation or part of the training population, yielding 10 accuracy estimates for every validation population. The final accuracy measure is the average of all the 10 accuracy measures (Equation 4).
Line 125: Equation (4) was not correctly displayed. “g_0” should be “g_0^hat”. There is a Chinese character on top of the equation. In addition, please use “cor” instead of “corr” for correlation.
ANSWER: The problem was caused by ‘save as pdf’. The word version is correct and so is the new pdf. Also, 'corr' was used in some of our previous publications, and should be left unchanged to be consistent.
Line 139-141: There are several errors in this statement. First, 5 different types of fourth-order epistasis times 7 traits give 35 heritability estimates, not 28. Second, I only observed 6 non-zero estimates ranged from 0.001 to 0.004, in which 5 for WT and one for TC.
ANSWER: All errors were corrected. Thanks.
Line 166: First of all, there were nine models (not seven) evaluated in Table S1. Secondly, what I understood is that the initial model listed in Table 1 was used as an “upper bound”. But still, not every possible combination smaller than the initial model was evaluated (e.g. A+D+AA+AD+DD+AAA). So how did you determine the models for evaluation?
ANSWER:
The seven models refer to those with epistasis effects. The other two models (A+D, A) were single-SNP models. A+D+AA+AD+DD+AAA was not included in Table S1 because none of the 3rd-order epistasis effects (AAA and AAD) with heritability >0.01 added to the prediction accuracy over the A+D+AD or the A+D+DD model, as indicated by the last model in Table S1 (A+D+AA+AD+DD+AAA+AAD). Had the last model increased the prediction accuracy over the previous models, two separate models would have been evaluated, one with AAA and one with AAD.
The very beginning of the original line 167 says: “The comparison of seven models with epistasis effects for WT showed the model with A×A×A and A×A×D had the same observed prediction accuracy as the five models with pairwise epistasis effects with observed prediction accuracy of 0.325, a 0.62% increase over the A+D SNP model (Table S1).”
Line 170-173: But both A+AD and A+AD+DD were not listed in Table 1. On the other hand, the accuracy of A+D+AA is 0.324, and A+D+AA+DD is 0.325. So, it seems that one cannot conclude that DD has no contribution.
ANSWER: Table 1 only lists the initial models that were subjected to accuracy evaluations after excluding effects with heritability <0.01. of each effect type under one model only: the full model with all 14 types of effects. Table 1 has no recommendation for the best model, which was decided by the best prediction accuracy.
The seven models refer to those with epistasis effects. The other two models were single-SNP models. A+D+AA+AD+DD+AAA was missed and now added. Prediction accuracy was the final judgement to include or exclude an effect types.
Line 174-176: For BMIo, the model A+D+AA+AD+DD was evaluated (Table S2), but it exceeds the corresponding initial model listed in Table 1, which is A+D+AA+AD. So, the same question as for line 166: how did you determine the models for evaluation? And the same for the trait HDL.
ANSWER: Table 1 does not list every model with better accuracy than the initial model. The sentence is now changed to: “… indicating that D×D effects had the same contribution to the observed prediction accuracy of BMIo as the A×D (Table S2). Therefore, either A×D or D×D could be included in the model but including both were unnecessary, and we used the A+AD model for BMIo.
Line 179-183: How was the best prediction models for the other four traits determined? In Tables S1-S3, you only showed the results of three traits.
ANSWER: The other four traits only had AxA with heritability > 1% that was included in the final model, This was stated as: ‘For four traits (LDL, TC, TG and HTo), A×A was the only epistasis effect with heritability greater than the 0.01 threshold, ranging from 0.135 for TG to 0.423 for LDL. Therefore, the prediction model for these four traits was A+D+AA.’ Lines 148-151.
Line 186-188: I would not call it a “substantial increase”. Even for the trait LDL with the largest relative increase (3.85%), the absolute increase of the mean accuracy was less than 0.01.
ANSWER: The sentence as revised as: “Of these seven traits, epistasis effects achieved accuracy increases over the SNP model for predicting the phenotypic values for six of the seven traits. Among these seven traits, LDL and HDL had the largest accuracy increases of 3.85% and 2.41% respectively, and four traits had minor accuracy increases for four traits, TC, HTo, WT and BMIo, with accuracy increases of 0.70%, 0.97%, 0.62% and 0.93% respectively. However, epistasis effects (A×A) decreased the prediction accuracy for TG (−2.35%) for unknown reasons”, lines 193-198.
In Table 2 and Figure 1, I would suggest write “global epistasis model” instead of “epistasis model”.
ANSWER: Accepted.
Line 244-246: The difference between the prediction accuracies was actually quite small, which I think is not enough to make such a strong conclusion. On the other hand, the heritability estimates in Table 3 indeed indicated that the local high-order epistasis contributed more than the global low-order epistasis to the genetic variance. There is still one point which has to be considered: The covariance matrices for the different types of genetic effects could be correlated. Thus, the relative contribution of different types of genetic effects may not be reliably inferred from the variance components (as well as the heritability estimates).
ANSWER: The last sentence of Conclusions is now revised as: “These results indicated that local high-order epistasis effects in haplotypes had a tendency to be more important than global pairwise epistasis effects for the seven human phenotypes”. Same changes were made in the abstract.
Reviewer 2 Report
Dear Authors
thank you for the manuscript - please see my comments below
my main question is if you could detail what you results mean for understanding the phenotypes; these are all measurements relevant for understanding cardiovascular diseases, how do your results help us understand these phenotypes, e.g. do they say something about the molecular mechanisms behind the phenotypes ?
- abstract line 21: please explain the AxD term
- abstract line 23: the last sentence is mangled
- Intro line 38 - 42: the sentence starting with "The epistasis effects contained ... "
is incomprehensible; what does the number of SNPs have to do with the ability to model and estimate epistasis effects ?
- Intro line 49: " not feasible to study ... " why ?
- Intro line 58 and 64 : "in" the EPIHAP program, not "by"
- line 70 : is there are reference for the data in the FHS that you used ?
- line 100 : please give an example relevant for the data and the study, not cattle
- Eq (1), (2), (3) : I think it would be helpful if you would give an actual example of how these objects would look like just for a minimal working example, say two phenotypes and five individuals, something like that
- Eq (4) is mangled
Author Response
We thank the reviewer for a very helpful review.
Texts in yellow are in response to the reviewer comments, and texts in green are previous texts answering some reviewer questions.
my main question is if you could detail what you results mean for understanding the phenotypes; these are all measurements relevant for understanding cardiovascular diseases, how do your results help us understand these phenotypes, e.g. do they say something about the molecular mechanisms behind the phenotypes ?
ANSWER: The main findings are summarized in the Abstract and Conclusions. Some answers to specific comments also have some answers to this comment.
- abstract line 21: please explain the AxD term
ANSWER: all effects are defined in the footnote of Table 1. Repeating those definitions in the main text could make the text look busy.
- abstract line 23: the last sentence is mangled
ANSWER: Problem fixed. Thanks.
- Intro line 38 - 42: the sentence starting with "The epistasis effects contained ... "is incomprehensible; what does the number of SNPs have to do with the ability to model and estimate epistasis effects ?
ANSWER: Line 27 defines epistasis effects as: “Epistasis effects are gene interaction effects and global epistasis effects of single nucleotide polymorphism (SNP) markers are interaction effects of genome-wide SNPs”. high-order epistasis effects involve more than two loci, and hence the number of SNPs is relevant. With 12 SNPs or more per haplotype, the epistasis effects in haplotypes could involve more than two loci, “although the exact number of SNPs involved in those high-order epistasis effects were unknown” (lines 40-41) . This study provided evidence “local high-order epistasis effects in haplotypes had a tendency to be more important than global pairwise epistasis effects for the seven human phenotypes”, which is the last sentence of the Conclusions, because haplotypes were more accurate than global pairwise epistasis effect.
- Intro line 49: " not feasible to study ... " why ?
ANSWER: This revisions adds a practical rather than theoretical answer: “as shown by the results of third- and fourth-order epistasis effects in this study” (line 49). The evidence from this study should contribute to the assessment of the feasibility of higher-order global epistasis effects.
- Intro line 58 and 64 : "in" the EPIHAP program, not "by"
ANSWER: Accepted.
- line 70 : is there are reference for the data in the FHS that you used ?
ANSWER: Yes, ref #8, line 74.
- line 100 : please give an example relevant for the data and the study, not cattle
ANSWER: Great point, changed as suggested. Thanks. A new sentence is added to describe the fixed effects: “Fixed effects included sex and cholesterol treatment as classification variables, and age, glucose and BMIo as covariables for HDL,TC, LDL and TG; and sex as classification variable and age as covariable for HTO, WT and BMIO.” (lines 106-109)
- Eq (1), (2), (3) : I think it would be helpful if you would give an actual example of how these objects would look like just for a minimal working example, say two phenotypes and five individuals, something like that
ANSWER: Ref#11 has a very detailed demonstration as “Tex S2. Numerical Demonstration’” in the supplementary material.
- Eq (4) is mangled
ANSWER: Problem was caused by ‘save as pdf’. Word version was correct.
Round 2
Reviewer 1 Report
The authors have not understood my point about cross-validation. The genuine 10-fold cross validation should be done as following: For each fold as test set, the trait values are predicted by the remaining nine folds as training set (as in this study). But, do NOT calculate the correlation between the predicted and observed values within each fold. Rather, one should pool the predictions of the 10-folds together and then calculate the correlation between the predicted and observed values for the entire data sets. If you are not familiar with this, just think about the extreme case: leave-one-out cross-validation.
As I said, the cross-validation which the authors did in this study is also acceptable because the poluation size in this study is large. Therefore, I do not request the authors to re-implement the genuine cross-validation. But the authors should note that their current implementation might give biased results in small populations (e.g. with less than 200 individuals).
Author Response
The reviewer made a clarification of a previous comment, suggesting that only one correlation should be calculated from the pooled results of all validation populations, but did not request any new change.
The reviewer's new clarification obviously forgot an important function of the cross validations: observe the variations among the validation populations using the standard deviation of the accuracy, which is shown as error bar in Figure 1. The standard deviation of the heritability estimates is included in Table S4.
Although the reviewer did not request new change, to remind this important function of cross validations, I added "Each error bar was ±1 standard deviation of the prediction accuracies of the ten validation populations" at the end of the title of Figure 1, lines 212-213 highlighted in yellow.